# Therapeutic Effect of Boron Neutron Capture Therapy on Boronophenylalanine Administration via Cerebrospinal Fluid Circulation in Glioma Rat Models

**DOI:** 10.3390/cells13191610

**Published:** 2024-09-25

**Authors:** Sachie Kusaka, Nikolaos Voulgaris, Kazuki Onishi, Junpei Ueda, Shigeyoshi Saito, Shingo Tamaki, Isao Murata, Takushi Takata, Minoru Suzuki

**Affiliations:** 1Division of Sustainable Energy and Environmental Engineering, Graduate School of Engineering, Osaka University, Yamadaoka 2-1, Suita 565-0871, Japan; voulgaris.nikolaos@qr.see.eng.osaka-u.ac.jp (N.V.); tamaki@see.eng.osaka-u.ac.jp (S.T.); murata@see.eng.osaka-u.ac.jp (I.M.); 2Division of Health Sciences, Graduate School of Medicine, Osaka University, Yamadaoka 2-1, Suita 565-0871, Japan; u677161e@ecs.osaka-u.ac.jp (K.O.); uedaj@sahs.med.osaka-u.ac.jp (J.U.); saito@sahs.med.osaka-u.ac.jp (S.S.); 3Institute for Integrated Radiation and Nuclear Science, Kyoto University, 2 Asashiro-Nishi, Kumatori-cho, Sennan-gun 590-0494, Japan; takata.takushi.6x@kyoto-u.ac.jp (T.T.); suzuki.minoru.3x@kyoto-u.ac.jp (M.S.)

**Keywords:** boron neutron capture therapy (BNCT), cerebrospinal fluid (CSF), boron delivery, brain tumor, glioma model rat, blood–brain barrier (BBB)

## Abstract

In recent years, various drug delivery systems circumventing the blood–brain barrier have emerged for treating brain tumors. This study aimed to improve the efficacy of brain tumor treatment in boron neutron capture therapy (BNCT) using cerebrospinal fluid (CSF) circulation to deliver boronophenylalanine (BPA) to targeted tumors. Previous experiments have demonstrated that boron accumulation in the brain cells of normal rats remains comparable to that after intravenous (IV) administration, despite BPA being administered via CSF at significantly lower doses (approximately 1/90 of IV doses). Based on these findings, BNCT was conducted on glioma model rats at the Kyoto University Research Reactor Institute (KUR), with BPA administered via CSF. This method involved implanting C6 cells into the brains of 8-week-old Wistar rats, followed by administering BPA and neutron irradiation after a 10-day period. In this study, the rats were divided into four groups: one receiving CSF administration, another receiving IV administration, and two control groups without BPA administration, with one subjected to neutron irradiation and the other not. In the CSF administration group, BPA was infused from the cisterna magna at 8 mg/kg/h for 2 h, while in the IV administration group, BPA was intravenously administered at 350 mg/kg via the tail vein over 1.5 h. Thermal neutron irradiation (5 MW) for 20 min, with an average fluence of 3.8 × 10^12^/cm^2^, was conducted at KUR’s heavy water neutron irradiation facility. Subsequently, all of the rats were monitored under identical conditions for 7 days, with pre- and post-irradiation tumor size assessed through MRI and pathological examination. The results indicate a remarkable therapeutic efficacy in both BPA-administered groups (CSF and IV). Notably, the rats treated with CSF administration exhibited diminished BPA accumulation in normal tissue compared to those treated with IV administration, alongside maintaining excellent overall health. Thus, CSF-based BPA administration holds promise as a novel drug delivery mechanism in BNCT.

## 1. Introduction

Glioblastoma (GBM) is the most common malignant intracranial tumor, with a low survival rate [1,2]. Despite treatment combining surgery and adjuvant radiochemotherapy, patients’ outcomes remain poor, with a 5-year overall survival rate of 5.4% [3]. Few therapies are effective for GBM due to distinct factors, including tumor invasiveness, an immunosuppressive microenvironment, adaptive resistance to therapy, and intratumoral heterogeneity [4,5].

Boron neutron capture therapy (BNCT) has recently garnered attention as a radiotherapy modality with few side effects [6,7,8]. The treatment involves the administration of a boron drug to a patient, as BNCT is a radiation therapy with neutrons [9]. An important feature of BNCT is that accumulated non-radioactive boron (^10^B) in cells induces a neutron nuclear reaction within subcellular compartments (10 µm), leading to cell death. This characteristic, known as “cell-selective particle therapy”, cannot be achieved with other radiotherapy methods. The ability of BNCT to minimize damage to normal brain cells while potentially treating infiltrating brain tumors with invisible borders is particularly noteworthy [10,11]; however, delivering sufficient ^10^B to brain tumor cells, which greatly influences therapeutic efficacy, remains challenging.

To ensure the treatment efficacy of BNCT, it is crucial that adequate ^10^B is concentrated in tumors in order to capture neutrons, trigger ^10^B (n,α)^7^Li reactions, and effectively kill cancer cells. This is especially important when targeting tumors such as glioma, which are protected by the blood–brain barrier (BBB) [12]. The BBB is integral in maintaining brain function, as it selectively restricts the passage of substances, which often hinders drug delivery to brain cells [13,14,15]. Moreover, reports suggest that the BBB may not be sufficiently compromised in GBM patients for adequate penetration of treatment agents [16,17,18]. Effective delivery strategies are therefore highly sought after to apply BNCT to brain tumor patients [19]. Recent advances have led to the development of several systems for drug delivery that bypass the BBB for brain tumor therapy [20,21,22]. Our laboratory has developed a unique method for delivering a boron-containing drug to brain cells. This method utilizes cerebrospinal fluid (CSF) circulation, as its application is aimed for use in BNCT. It has been named the “boron CSF administration method” (Figure 1). Our understanding of CSF circulation has evolved since the discovery of meningeal lymphatic vessels in 2015 [23]. It is recognized that the glymphatic system, which is a major pathway for CSF clearance, facilitates bulk flow via the spinal and cranial nerves, enabling the transport of therapeutic molecules to deep brain regions via the CSF microcirculation mechanism [24].

In our previous studies, important results have been obtained from various experiments based on the CSF administration method using boronophenylalanine (BPA), a widely used boron drug in BNCT. In the initial study, using rats without transplanted tumors, we found that the boron CSF administration method achieved brain cell boron accumulation equivalent to the intravenous (IV) administration method despite employing a significantly lower BPA dose (approximately 1/90 of the IV method dose) [25]. Continuous BPA infusion into the CSF for at least 60 min was necessary to saturate the boron concentration in the CSF, and in BPA-saturated CSF, boron uptake into brain cells was shown to increase slightly with increasing BPA dosage, but with little correlation observed. Additionally, melanoma model rats administered BPA via the CSF method exhibited a high T/N ratio (the ratio of boron concentration in tumor cells to normal cells) [26]. In our latest experiment, BPA accumulation in the brain parenchyma was visually confirmed in mass spectrometry imaging analysis after over 60 min of CSF administration, and it was also observed that cessation of BPA administration resulted in swift excretion of BPA from the brain parenchyma [27]. These results suggested that BPA uptake into the brain parenchyma may be influenced by the CSF compartment’s positive pressure during and after drug administration.

As mentioned above, CSF administration of BPA resulted in the efficient accumulation of tumors even with low doses, as well as swift excretion from normal brain tissue. Based on these findings, we hypothesized that the CSF administration method could achieve equivalent or superior therapeutic effects compared to IV administration, even with a low BPA dose. To demonstrate this hypothesis, in the present experiment, we designed a CSF-based administration protocol of BNCT for rat glioma models, and a thermal neutron irradiation experiment was conducted at the Kyoto University Research Reactor (KUR).

## 2. Materials and Methods

The protocol of the experiments in Section 2.1–Section 2.3 was approved by the Animal Care and Use Committee for Osaka University (approval number 2023-2-1) and the Institute for Integrated Radiation and Nuclear Science, Kyoto University (approval number 2023-16).

### 2.1. C6 Glioma Model Orthotopic Rats

In this study, eight-week-old male Wistar rats, each weighing approximately 180 to 200 g, were utilized, sourced from Japan SLC, Inc., Shizuoka, Japan. The C6 tumor cells were obtained from Tohoku University and implanted into the rats’ brains using the following procedure: A total of 4.0 × 10^5^ cells/5 μL was prepared and injected at a rate of 1 μL per minute over a period of 5 min. The injection site was located 4 mm to the right of the bregma, 0 mm caudal, and 4 mm ventral.

### 2.2. BNCT Effect on the CSF Administration Method of ^10^BPA

^10^BPA was generously provided by Itsuro Kato [28] and prepared as a fructose complex to enhance its water solubility (20 mg/mL BPA). The C6 rat glioma models were supplied for this experiment 10 days after implantation. The thermal neutron irradiation experiment was conducted at KUR (Institute for Integrated Radiation and Nuclear Science, Kyoto University, Osaka, Japan).

Sixteen rat tumor models were randomly divided into four groups with four rats per group: an untreated control group (non-irradiation), a neutron-irradiated control group (irradiation only), and one group subjected to thermal neutron irradiation after the end of BPA infusion via the IV route and another via the CSF route.

In the IV administration group, four rats received 350 mg/kg of BPA via the tail vein over 1.5 h. In the CSF administration group, BPA was administered to four rats via the intracisterna magna at a rate of 8.0 mg/kg/h for 2 h. All rats were anesthetized via an intraperitoneal injection of a mixture of anesthetics, including medetomidine (Nippon Zenyaku Kogyo Co., Ltd., Fukushima, Japan) (0.15 mg/kg), midazolam (SANDOZ, Tokyo, Japan) (2.0 mg/kg), and butorphanol (Meiji Animal Health Co., Ltd., Tokyo, Japan) (2.5 mg/kg). These experimental conditions are summarized in Table 1.

During radiation treatment at KUR, all rat bodies, except for their heads, were attached to a plate lined with ^6^LiF ceramic tiles to shield against and reduce neutron irradiation before the thermal neutron irradiation was performed. The C6 glioma rats were irradiated with a 5 MW reactor at a heavy water irradiation facility for 20 min (average of 3.8 × 10^12^ neutrons/cm^2^). Following thermal neutron irradiation, all rats remained under the same experimental conditions as the control groups for seven days. The entire heads of the rats, including their brains, were fixed in 10% paraformaldehyde for post-treatment MRI evaluation and histological evaluation of the glioma and their brains.

### 2.3. Pre- and Post-Treatment MRI Assessment

An ultrahigh-field 7-Tesla (7-T) MRI system was used to obtain the MRI scans. This system has a ^1^H volume transmit–receive coil (PharmaScan^®^ 7T; Bruker, Ettlingen, Germany). For the in vivo axial T2-weighted images (T2WIs), the turbo RARE (rapid acquisition with the relaxation enhancement) sequence was employed. The parameters set were as follows: (repetition time)/(echo time) ratio: (TR)/(TE) = 3200/33 ms; 20 slices; a RARE factor of 8; a field of view of 32.0 × 32.0 mm^2^; 4 averages; a matrix size of 256 × 256; a slice thickness of 1.0 mm; and a total scan time of 6 min and 50 s. The ex vivo axial T2WIs were obtained using the turbo rapid acquisition with the relaxation enhancement sequence. The following Turbo RARE sequence parameters were used: (repetition time) (echo time): (TR)/(TE) = 3200/33 ms; 20 slices; a RARE factor of 8; a field of view of 32.0 × 32.0 mm^2^; 8 averages; a matrix size of 256 × 256; a slice thickness of 1.0 mm; and a total scan time of 13 min 39 s [29].

Pre-treatment MRI scans of the C6 rat glioma models were performed 3 days before the irradiation experiment (BNCT). The three-day gap between the MRI scan and BNCT was due to the time required for the MRI scan at Osaka University and the transportation of the rats to KUR. If a given neutron irradiation facility has diagnostic imaging equipment, it would be preferable to conduct the scans immediately before BNCT. Post-treatment MRI scans were conducted 7 days after irradiation. The MR images were analyzed using ImageJ software (version 1.54i, https://imagej.net/, accessed on 24 September 2024). Regions of interest (ROIs) were manually delineated within the tumor region of the brain section. Tumor volume was calculated by summing the areas for each slice of the MR image (slice thickness = 1.0 mm), and the average volume value determined by two observers was calculated for each sample.

### 2.4. Hematoxylin and Eosin Staining of C6 Rat Glioma Brain Sections

After approximately one month of decalcification of the entire heads of the rats, standard paraffin-embedded tissue sections were prepared and placed on slides that were stained with hematoxylin and eosin (HE) using standard histological procedures (Biopathology Institute Co., Ltd., Oita, Japan). All HE-stained slides were observed using a fluorescent microscope (BZ-X810; KEYENCE CORPORATION, Osaka, Japan).

### 2.5. Boron Concentrations in Various Normal Tissues of Rat Heads Administered BPA via Both the CSF and IV Methods

As a reference for BNCT in Section 2.2, we measured the boron concentration in normal tissues of rat heads without tumor implantation. First, BPA was administered following the same protocol as the above experiments. In this experiment, BPA was administered via the cisterna magna in 5 rats and via the tail vein in 5 rats. After administration, the buccal mucosa, tongue, eyeball, and brain were promptly collected from each individual. The collected samples were stored at −80 °C until the boron concentration was measured using inductively coupled plasma–atomic emission spectrometry (ICP–AES).

### 2.6. Statistics

The estimated parameter values, including tumor volume, are presented as means ± SDs. Differences between groups in the estimated parameters were evaluated using a one-way ANOVA followed by Dunnett’s test, with the analysis conducted in Prism 10 (Version 10.2.3; GraphPad Software, San Diego, CA, USA).

## 3. Results

In this study, we investigated whether administering BPA via CSF in BNCT is effective at therapeutic levels even with lower dosages. Using MRI and pathological examination, we compared tumor volumes and therapeutic efficacies before and after BNCT in both IV- and CSF-based methods. Figure 2 illustrates the tumor volume ratio before and after BNCT for each group. As shown in Figure 2, significant therapeutic effects are observed in both the IV and CSF administration groups compared to the control groups.

**Figure 2 cells-13-01610-f002:**
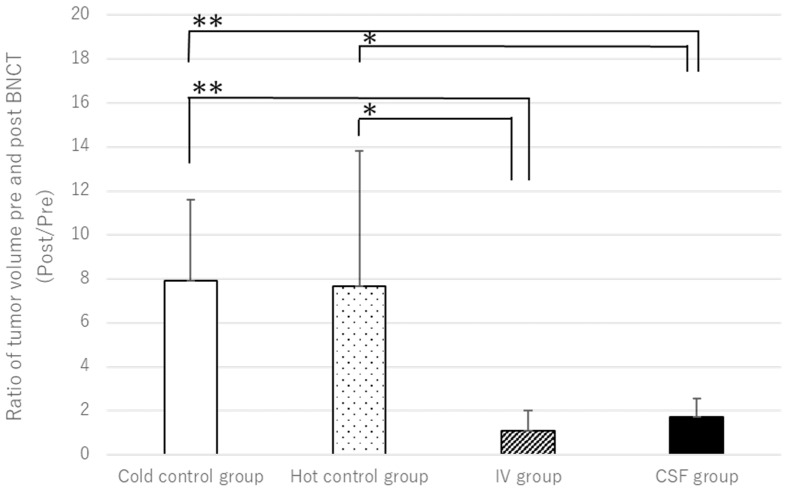
The figure illustrates the ratio of tumor volumes before and after BNCT for each group. Compared to the control groups (A′/A and B′/B), both the CSF (D′/D) and IV (C′/C) groups exhibit a suppressed increase in tumor size (**: *p* < 0.01, *: *p* < 0.1).

The MR images for each group are depicted in Figure 3. Compared to the control groups, inhibition of tumor growth is observed in the groups administered BPA (via both IV and CSF).

**Figure 3 cells-13-01610-f003:**
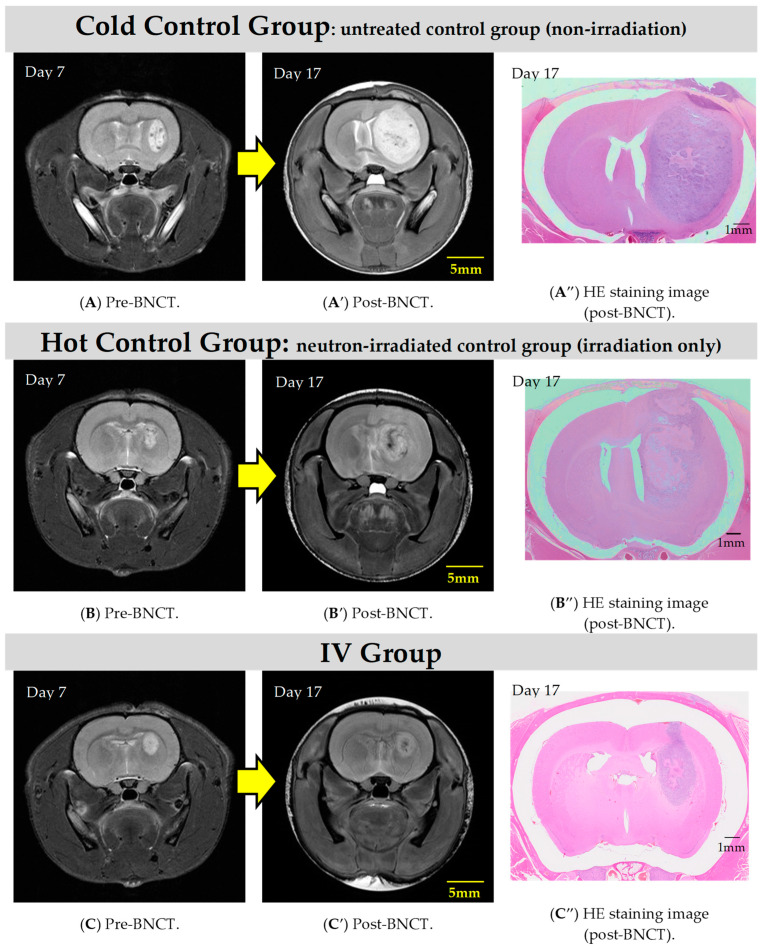
The T2WIs show representative examples of rat heads before and after BNCT in each group. It can be observed that increases in tumor size are suppressed in the IV (**C**,**C′**,**C″**) and CSF (**D_1_**–**D_4_**,**D_1_′–D_4_′**,**D_1_″**–**D_4_″**) groups compared to the control groups (cold (**A**,**A′**,**A″**) and hot (**B**,**B′**,**B″**)).

In the present study, some rats exhibit not only inhibited tumor growth but also a significant reduction in tumor volume as early as one week after BNCT with the IV BPA administration, highlighting the notable effectiveness of this therapy in rat models. Figure 4 demonstrates the remarkable histologically normal tissue levels post-treatment. When comparing the two administration methods, based on pathological examination, the IV method with a BPA dose of 350 mg/kg demonstrates better performance over the CSF method with a dose of 8 mg/kg/h (total 16 mg/kg), albeit only slightly. 

However, all rats in the IV method group experienced significant weight loss (Figure 5) and exhibited considerable debilitation one week after BNCT. Conversely, the CSF method group showed a relatively closer resemblance to the control group, with weight recovery observed.

Figure 6 shows the concentration of boron accumulation in normal tissues of rat heads following the infusion of BPA via the tail vein and cisterna magna under conditions identical to those of the BNCT in this study. Particularly noteworthy is the significant boron accumulation observed in oral tissues with the IV administration of BPA, whereas CSF administration results in comparatively lower boron accumulation in normal tissues.

## 4. Discussion

In this study, we demonstrated that harnessing CSF circulation for BPA is likely to be an efficient method for administering boron drugs in BNCT. Tumor growth was suppressed more effectively in the CSF group than in the control groups and was similar to the IV group. Moreover, the IV group required a significantly higher dose of BPA, leading to worse general conditions in rats due to side effects on normal cells. In contrast, there were minimal impacts on normal cells in the CSF group, indicating that CSF administration of BPA is a biologically friendlier method with a superior performance to IV administration.

### 4.1. Why Does BPA Accumulate in Brain Tumors Despite Small Doses of CSF

After substances are exchanged with brain interstitial fluid, they become associated with the efferent paravascular glial lymphatic (glymphatic) system (perivascular space on the venous side) that carries the CSF, as well as the recently described meningeal lymphatic system [30]. It has recently been discovered that solutes may flow into the brain parenchyma, with astrocyte expression of aquaporin 4 (AQP4) playing an important role in regulating perivascular CSF inflow and outflow (via the glymphatic pathway) [31,32,33]. This flow (bulk flow) generated by AQP4, as shown in Figure 7, is essential for removing substances no longer needed by brain cells into the CSF [34]. Based on these findings, we hypothesized that by administering BPA via CSF, BPA can be delivered directly from the CSF to the brain interstitial fluid and then to brain tumors, even though the dose is low.

However, it remains unclear within the scope of this study whether this pathway is dependent on the pressure in the CSF compartment, the drug concentration in the CSF or brain tissue, or other factors. Assuming that drug administration from the CSF is influenced by cerebral pressure or drug concentration, predicting the relationship between the dose and amount accumulated in the brain is not only complicated, but the administration of drugs that have side effects at very small doses may also be dangerous. There are several limitations to the types of drugs that can be administered via CSF, including concerns about intracranial pressure and neurotoxicity. On the other hand, in BNCT, BPA commonly used via the IV route has been proven to have minimal side effects, even when administered in doses as high as 30 g/60 kg of adult human body weight. In other words, BPA can be considered a drug with a wide safety margin, which suggests that it may be a promising candidate for administration via CSF in BNCT. Furthermore, due to concerns about toxicity and cost, many chemical compounds that have been studied with the assumption of IV administration, yet have not advanced in drug development, may have potential for use via the CSF route because the required dosage is smaller. In future research, it is necessary to consider whether the irradiation protocol currently used in clinical practice, which involves the continuous infusion of boron drugs, can also be applied to CSF-based administration. Additionally, emphasis on the field of neuroscience will be crucial to elucidate the specific factors involved in enabling BPA administered via CSF to deliver large amounts of boron to tumors.

### 4.2. Boron Concentration in Normal Tissues and the T/N Ratio in BNCT

In our study, we observed significant tumor regression in several rats who received IV administration of BPA; however, we noted a notable decrease in body weight one week after irradiation in this group (Figure 5), indicating an overall decline in health. Figure 6 illustrates the concentration of boron accumulation in normal tissues of rat heads following the infusion of BPA via the tail vein and cisterna magna under conditions identical to those of our study. As shown in the figure, there was a substantial accumulation of boron in the oral mucosa and tongue, particularly in the IV infusion group. Based on previous experiments, it was suggested that the rats subjected to BNCT after BPA IV administration in our study may have experienced difficulty in food intake post-treatment due to potential adverse effects. Specifically, it is challenging to restrict neutron irradiation to the tumor alone in small animals, resulting in the irradiation of the entire head with neutrons. Consequently, radiation exposure to the radiation-sensitive oral mucosa leads to a decrease in quality of life (QOL). While limiting neutrons in human BNCT may partially mitigate oral mucosal exposure, the significant accumulation of boron in normal tissues needs to be considered [35].

Conversely, the accumulation of boron in normal tissues of the head was minimal in the case of CSF infusion, as depicted in Figure 6. Post-treatment, we observed an increase in body weight similar to that of the control group. The present CSF-based results showed that the reduction rate of brain tumors (relative tumor volume) was similar to that of IV administration. What deserves attention here is that the phenomenon could result in an increase in the T/N ratio. In order to achieve therapeutic efficacy in BNCT, not only the boron amount in tumor cells but also the T/N ratio is crucial. In this study, despite the very low dose of BPA administered via CSF, results comparable to those of the IV method were achieved while maintaining the overall good condition of the rats. This suggests that CSF administration can offer a significant advantage in achieving a high T/N ratio for BNCT, as it results not only in a relatively high amount of boron delivered to the brain tumor but also in minimal boron accumulation in normal tissues.

### 4.3. BPA Pharmacokinetics and Pharmacodynamics in Brain Tissue

In our previous study, we investigated the pharmacokinetics of BPA in brain tissue by administering BPA via IV or CSF for 1 h in normal rats. After BPA infusion into the CSF was stopped, it was shown that the boron concentration in both the CSF compartment and normal brain cells decreased rapidly (Figure 8) [26]. This phenomenon can be easily explained by the fact that the primary function of the CSF is to facilitate drainage from the brain. On the other hand, BPA remained in the brain tissue for an extended period after IV administration. These results strongly support the findings regarding rats’ body weight changes in normal tissues discussed in Section 4.2. Additionally, we previously confirmed that BPA administered at 4 mg/kg/h via CSF in brain tumor model rats (melanoma) resulted in a similar level of boron accumulated in tumors to IV administration at 350 mg/kg/h [26]. This result is consistent with the pharmacodynamic analysis in the current experiment. From these findings, it can be concluded that BPA administration via CSF can achieve therapeutic effects comparable to those of IV administration with a lower dose and that rapid excretion of BPA from the CSF may result in a significantly higher T/N ratio for CSF administration.

## 5. Conclusions

In this study, we investigated whether administering a small amount of BPA into the CSF could achieve therapeutic effects equivalent to those achieved via IV administration. CSF administration not only demonstrated this effect, but the overall condition of the rats was also good. It also has the advantage of being used in smaller amounts. Due to toxicity concerns, chemical compounds studied for IV administration that have not progressed in drug development might be viable for CSF administration because they require a smaller dosage. Furthermore, CSF administration can significantly improve the T/N ratio for BNCT by delivering the required boron to brain tumors and reducing accumulation in normal tissues. However, there are several limitations to CSF administration, and many factors affecting this route remain poorly understood. The knowledge of CSF circulation is increasing year by year, and, in the future, this administration method may be crucial in the application of BNCT to brain tumors. Simultaneously, further research on CSF circulation is necessary. In future studies, BNCT via the CSF will be conducted using drugs containing higher concentrations of boron than BPA, and it will be necessary to further investigate which factors, including brain pressure and boron concentration in the CSF, increase delivery to brain tumors.

## Figures and Tables

**Figure 1 cells-13-01610-f001:**
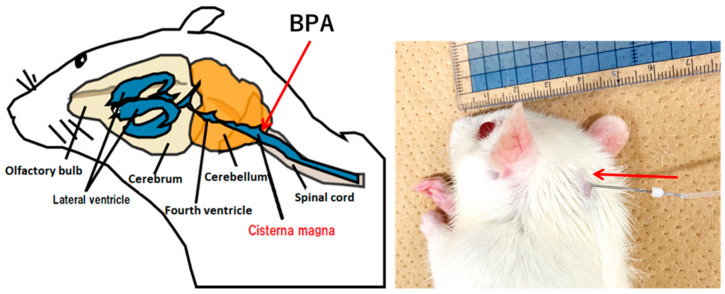
Photograph of boron CSF administration method in a rat (**right**) and schematic illustration (**left**).

**Figure 4 cells-13-01610-f004:**
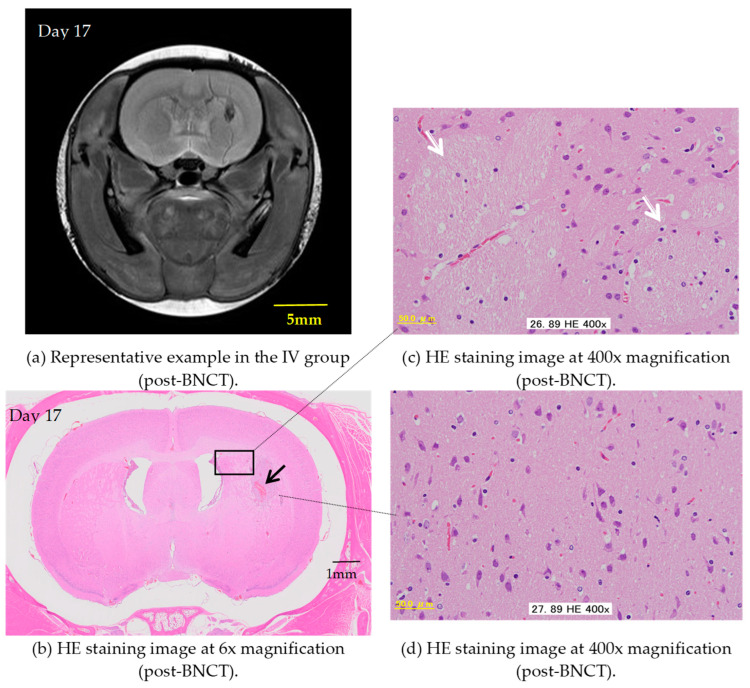
The figures represent an example of the IV group. (**a**) T2WIs after BNCT. (**b**–**d**) Show thin sections of rat brain tissue stained with HE depicting the brain’s condition after BNCT with the IV administration of BPA. (**b**) Shows 6× magnification under a microscope. Some hemorrhaging (⟶) is observed in the striatal area, but there is minimal evidence of inflammatory cell infiltration. (**c**) Shows 400× magnification under a microscope (the area enclosed by the rectangle in Figure (**b**)). The image indicates that the fiber bundles in the cerebral cortex near the hemorrhagic lesion (⇨) are at normal levels. (**d**) Shows 400× magnification under a microscope. Unlike the other samples, no signs of demyelination are observed in this specimen.

**Figure 5 cells-13-01610-f005:**
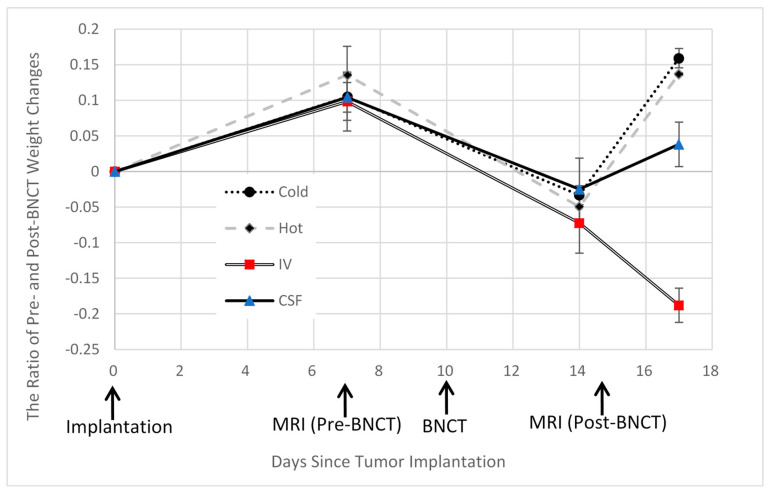
This figure illustrates the changes in the body weight of rats in each group, with the day of C6 cell transplantation considered as day 0. Following BNCT, all groups except the IV group exhibited weight recovery, whereas the IV group showed approximately 20% weight loss one week after treatment.

**Figure 6 cells-13-01610-f006:**
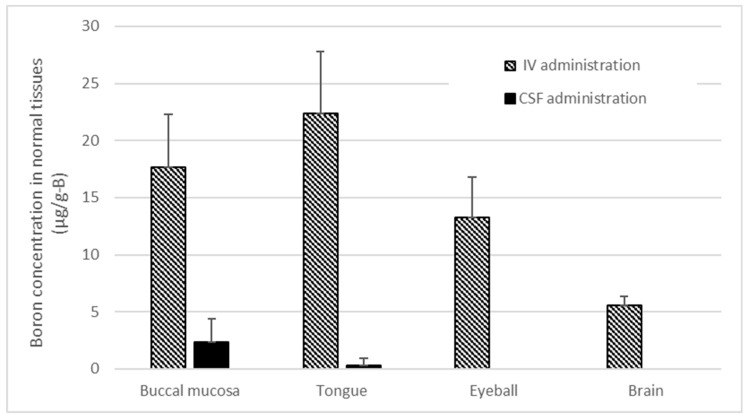
This figure shows the boron concentrations in various normal tissues of rat heads when BPA was administered via both the CSF and IV methods, following the same administration protocol as in the irradiation experiment.

**Figure 7 cells-13-01610-f007:**
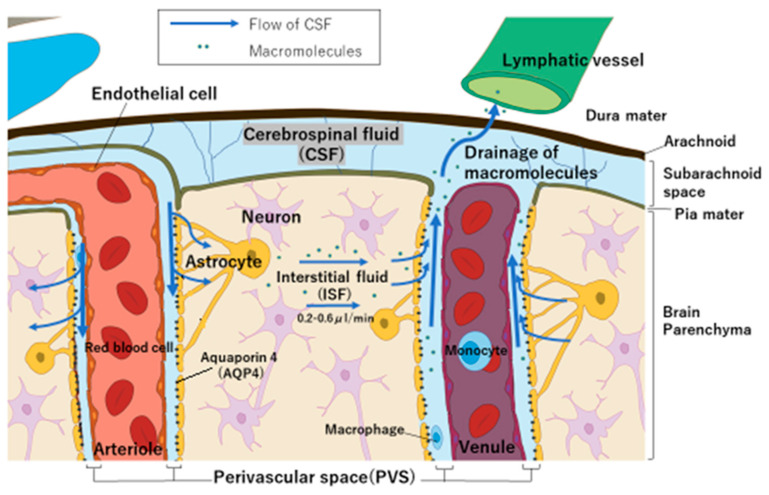
Illustration depicting the anatomy of the brain and meninges, including meningeal lymphatic vessels, cerebral arteries and veins, brain cell types, and the flow of CSF.

**Figure 8 cells-13-01610-f008:**
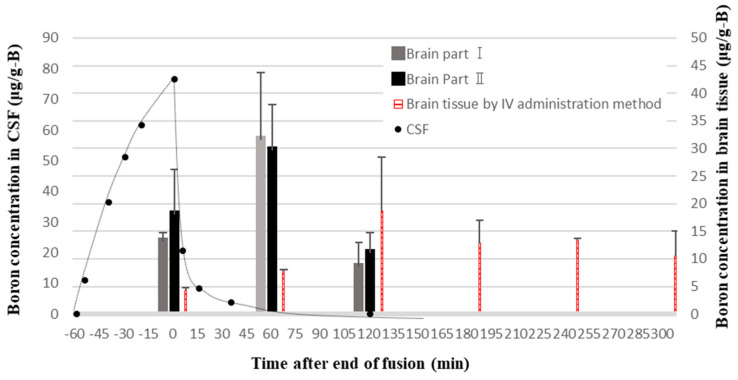
Excretion time profile of boron concentration in brain cells after BPA administration into the lateral ventricle or the tail vein of normal rats. The end of infusion is set to 0 min. Brain Part I: a portion of brain tissue located near the lateral ventricle; Brain Part II: a portion of brain tissue away from the lateral ventricle [26].

**Table 1 cells-13-01610-t001:** Schematic of the experiment and the title of each group as used in Figure 2 and Figure 3 (results).

	Day 0 *	Day 7	Day 10		Day 17	
	C6 Cell Transplantation	MRI (Pre-BNCT)	BNCT		MRI (Post-BNCT)	HE Staining
			BPA	Irradiation		
**Cold control group** **	◯	A	-	-	A′	A″
**Hot control group** ***	〇	B	-	20 min	B′	B″
**IV group**	〇	C	350 mg/kg	20 min	C′	C″
**CSF group**	〇	D	16 mg/kg	20 min	D′	D″

* C6 cell transplantation day as day 0. ** The untreated control group (non-irradiation) is defined as the cold control group. *** Neutron-irradiated control group (irradiation only) is defined as the hot control group.

## Data Availability

The original contributions presented in the study are included in the article, further inquiries can be directed to the corresponding authors.

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
