# Peer review of "Therapeutic Effect of Boron Neutron Capture Therapy on Boronophenylalanine Administration via Cerebrospinal Fluid Circulation in Glioma Rat Models"

_cells, 2024, doi:10.3390/cells13191610_

Round 1
Reviewer 1 Report
Comments and Suggestions for Authors
Kusaka et al have writeen an interesting work related to the therapeutic effect of Boron Neutron Capture Therapy (BNCT) on Boronophenylalanine (BPA) Administration via Cerebro Spinal Fluid (CSF) in Gloima rats. They could shown a good therapeutic efficacy for BPA in CSF as well as intravenous (IV) application.
These findings could be first implications as treatment options in glioma for BNCT. Small comments would be helpful for a better understanding:
1) How exactly was the CSF application performed? A picutre or model of inserting would be interesting?
2) Could the way of application be a reason for inferior results compare to IV group?
3) Authors should explain the reason of doing MRI postoperatively at 7th day.
4) Had the authors ever though about implantation of a device into the rats for the application?
All in all results and methods are interesting. Some of these minor recommendations should be addressed.
Author Response
Thank you very much for carefully reviewing and providing feedback on our paper. Please see the attachment.

Reviewer 2 Report
Comments and Suggestions for Authors
The manuscript demonstrates that intrathecal delivery of boronophenylalanine improves the therapeutic efficacy of boron neutron capture therapy in rat glioma models. The design and protocol are generally rational, and the results are sound. I have just a few minor comments for the authors to address:
-
The authors have previously reported on boron delivery to brain tumors for BNCT in melanoma models (doi: 10.3390/biology11030397), with a focus on drug kinetics in CSF. The results are quite similar to this study, particularly regarding the CSF dose, which is only 1/90 of that for IV administration. Ideally, these two studies could be combined into a comprehensive paper with both PK and PD analyses, but keeping them separate is also acceptable.
-
Although BNCT typically requires a single treatment session, a 2-hour BPA infusion via the intracisterna magna seems too long to be clinically practical. This could raise concerns about potential infection and poor patient compliance. Given that BPA is a small molecule with fast clearance through IV, IP, and CSF routes, a sustained release formulation of BPA might be viable. A discussion on alternative strategies to improve delivery efficiency should be included.
-
The goal of managing brain tumors is to extend overall survival, with standard therapies typically involving long-term treatment. In preclinical studies, a typical efficacy study for tumor-bearing animals spans around two months. Therefore, extending the observation period beyond 18 days post-tumor inoculation is recommended to provide more comprehensive insights.
Author Response

(The authors gave the same response as above.)

Reviewer 3 Report
Comments and Suggestions for Authors
I believe that the manuscript is very well written and covers the journal's range.
Author Response

(The authors gave the same response as above.)
